# PROMPT SEGMENTATION AND ANNOTATION OPTIMISATION: CONTROLLING LLM BEHAVIOUR VIA OPTIMISED SEGMENT-LEVEL ANNOTATIONS

## ABSTRACT

Prompt engineering is crucial for effective interaction with generative artificial intelligence systems, yet existing optimisation methods often operate over an unstructured and vast prompt space, leading to high computational costs and potential distortions of the original intent. We introduce Prompt Segmentation and Annotation Optimisation (PSAO), a lightweight and model-agnostic framework designed to improve prompt controllability and efficiency. PSAO decomposes a prompt into interpretable segments (e.g., sentences) and augments each with human-readable annotations (e.g., not important, important, very important). These annotations guide large language models (LLMs) to allocate attention more effectively during response generation. We formally define the segmentations and annotations and provide theoretical guarantees that PSAO yields responses that are provably at least as good as, and often better than, those generated from the original prompt. Empirical results demonstrate that PSAO enhances LLM performance and can be seamlessly integrated with existing prompt optimisation methods or used as a stand-alone approach.

## 1 INTRODUCTION

Large language models (LLMs) have demonstrated remarkable capabilities across diverse tasks such as summarisation, reasoning, and code generation (Hendrycks et al., 2021; Wang et al., 2024). However, their performance is highly sensitive to the wording of input prompts (Zhan et al., 2024; Tang et al., 2025). End-users interact with LLMs through prompting, making prompt engineering essential in optimising LLM performance. Recent studies have shown that even small prompt modifications can lead to significant performance differences (Huang et al., 2025; Liskavets et al., 2025; Trivedi et al., 2025).

To improve prompt quality, numerous automatic prompt optimisation methods have been proposed. For example, gradient-based (Wen et al., 2023; Pryzant et al., 2023) or gradient-free (Shin et al., 2020) search and scoring approaches, reinforcement learning-based optimisation (Deng et al., 2022; Li et al., 2024; Huang et al., 2025), and methods leveraging (LLMs) as optimisers (Tang et al., 2025), such as Automatic Prompt Engineering (APE) (Zhou et al., 2023b), which iteratively refine prompts.

While these approaches have significantly advanced the state of the art, they share several common limitations. Many operate in the vast and unstructured space of natural language, which can result in computational inefficiency and poor sample efficiency (Pryzant et al., 2023; Zhou et al., 2023a). Some methods risk semantic drift, where optimised prompts deviate from original inputs, resulting in reduced readability and limiting the user's ability to refine the prompt further using domain knowledge (Zhou et al., 2025). Furthermore, LLM-based optimisation methods often lack guarantees of exploring the optimal regions within the search space, underscoring the need for more principled and efficient optimisation strategies (Sabbatella et al., 2024).

In contrast, humans often process large or complex contexts by adding lightweight annotations, such as underlining key sentences, marking text as *important* or adding *explanations*. Such annotations help direct attention without altering the original text.

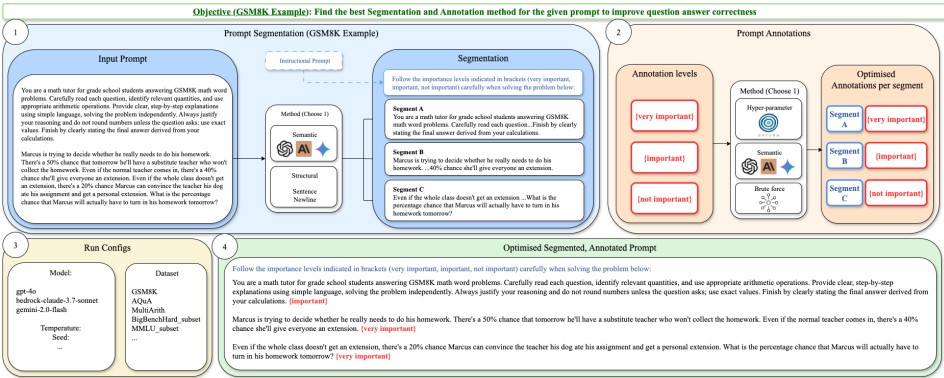

Figure 1: An illustration of Prompt Segmentation and Annotation optimisation (example: GSM8K question). Initially, a prompt is provided to PSAO, which is segmented into chunks (example: segmented by semantic meaning via an LLM). An instructional prompt describing the annotation may be added at the beginning. Next, PSAO assigns an annotation to each prompt segment to facilitate optimisation (example: the structured annotation is represented as {very important}, {important} and {not important}). Finally, the optimised prompt is generated by concatenating the segmented portions of the prompt with their corresponding optimised annotations, aiming to maximise LLM performance. Evaluation: annotations yielded a 50% uplift in accuracy (167% relative uplift) from 30% to 80%. This was determined by running the original prompt vs annotated prompt 20 times.

Inspired by this, we hypothesise that *structured or unstructured, human-readable annotations can serve as controllable signals for LLMs, improving reasoning and alignment while preserving semantic intent.*

We propose **Prompt Segmentation and Annotation Optimisation (PSAO)**, a novel and lightweight framework for controllable prompt optimisation. PSAO decomposes a prompt into interpretable segments (e.g., sentences or clauses) and attaches human-readable control annotations (e.g., {important}, {not important}, {very important}) to each segment. These annotations act as soft directives, guiding LLMs' attention and reasoning. By optimising annotation assignments rather than rewriting the prompt itself, PSAO preserves the prompt's original meaning while providing a structured and tractable optimisation space. In this paper, we formally define the prompt segmentation and annotation optimisation problem and establish four key theoretical guarantees. Empirical evaluations across multiple tasks demonstrate that PSAO can improve LLM performance with negligible computational overhead. PSAO is model-agnostic, interpretable, and easily applicable in real-world scenarios, offering a new perspective for controllable and efficient prompt optimisation.

The main contribution of this paper are as follows:

1. We introduce the Prompt Segmentation and Annotation optimisation (PSAO) framework, which enables more effective and interpretable prompt optimisation for LLMs.

2. We provide a comprehensive theoretical analysis of PSAO, establishing its key properties and guarantees.

3. We conduct empirical evaluations to demonstrate the potential of PSAO across various tasks and models.

## 2 RELATED WORK

Prompt optimisation refers to the process of refining, engineering or automatically generating prompts to improve LLM performance on downstream tasks (Zhou et al., 2023b; Yang et al., 2024; Sun et al., 2024; Tang et al., 2025). Methods span gradient-based soft prompt tuning to gradient-free strategies such as meta prompting and LLM-in-the-loop refinement.

Initial work explored discrete prompt optimisation through gradient-based search, including Auto-Prompt (Shin et al., 2020) and hard prompt tuning via gradient descent and beam search (Wen et al.,

2023). Prior work on soft prompt tuning optimised prompts as task-specific continuous embeddings prepended to model inputs (Lester et al., 2021; Li & Liang, 2021; Lan et al., 2025; Fan et al., 2025), with extensions such as gradient-based reasoning enhancements (Das et al., 2025). Other approaches use reinforcement learning and token-level editing to improve prompt quality, including RLPrompt (Deng et al., 2022), DP2O (Li et al., 2024), RTLIR (Huang et al., 2025), and ParetoPrompt (Zhao et al., 2025). These techniques require access to gradients, model weights or internal parameters, making them unsuitable for black-box or API-only settings (Chen et al., 2024; Yang et al., 2024).

In parallel, gradient-free methods increasingly rely on LLMs to generate and refine prompt candidates. APE (Zhou et al., 2023b) generates prompt variants from an initial instruction and selects those with the highest downstream performance. Recent methods incorporate feedback to guide revisions, such as OPRO (Yang et al., 2024), which uses a trajectory of prompt-score pairs to inform future edits and CriSPO (He et al., 2025), which adds structured critiques for multi-aspect self-reflection. In addition, ProTeGi (Pryzant et al., 2023) edits prompts using natural language critiques with beam and bandit-guided search, while GEPA (Agrawal et al., 2025) applies evolutionary refinement through Pareto-guided, alignment-driven reflection. This extends to output-level revision, with methods like Reflexion (Shinn et al., 2023) using verbal feedback on prior outputs, and approaches such as PromptAgent (Wang et al., 2024), Zhang et al. (2024) and Ye et al. (2024) adopt multi-round refinement through self-critique or planning-based strategies.

Several works explore programmatic and modular strategies for prompt optimisation. DSPy provides an interface for multi-stage pipeline optimisation (Khattab et al., 2024). Within this, COPRO uses coordinate ascent to refine prompts (Khattab et al., 2024; Sarmah et al., 2024), and MIPRO combines instruction and example selection with Bayesian optimisation (Opsahl-Ong et al., 2024). Related segmentation-based methods decompose tasks or prompts into sub-prompts or segment-level cues (Khot et al., 2023; Jain & Chowdhary, 2025). Some methods use language models as mutation and crossover operators in evolutionary algorithms, as in EvoPrompting (Chen et al., 2023), EvoPrompt (Guo et al., 2024) and PromptBreeder (Fernando et al., 2024). Causal prompting (Zhang et al., 2025) uses structural causal models to estimate prompt effects and guide optimisation.

Existing methods share several limitations: they often treat prompts as monolithic (Zhou et al., 2023b), involve extensive LLM queries (Shinn et al., 2023) and risk diverging from the user's original intent (Wu et al., 2024). The lack of interpretable structure further limits transparency and reduces users' ability to understand or control the optimisation process (Bie et al., 2024; Feng et al., 2024).

## 3  METHODOLOGY

### 3.1  PROBLEM SETTING

Given a natural language prompt $P \in \mathcal{L}$, where $\mathcal{L}$ denotes the space of all natural language prompts, and a frozen large language model $\mathcal{M}$, the goal of prompt optimisation is to identify an optimal prompt $P^*$ such that the generated output $\mathcal{M}(P^*)$ maximises a given objective function $Q(\bullet)$. The objective $Q(\bullet)$ can represent various criteria, including task accuracy, response coherence, informativeness, or human preference.

**Challenge:**  Most existing prompt optimisation methods operate directly within the natural language space $\mathcal{L}$, which poses several inherent challenges:

1. **Highly-dimensional and combinatorially large**, making exhaustive or fine-grained search intractable,

2. **Non-differentiable**, leading to inefficient optimisation procedures,

3. **Semantically unstable**, as optimisation may unintentionally alter the original intent of the prompt.

These limitations motivate us to explore alternative strategies for prompt optimisation. Inspired by how humans annotate long or complex contexts to convey relative importance, we propose leveraging *structured representations* and *intermediate annotations*. Such representations aim to (i) capture the essential semantics of the original prompt while being more amenable to optimisation, (ii) pre-

serve the readability and intent of the prompt, and (iii) provide meaningful guidance to the optimisation process through explicit annotations.

**To achieve the best LLM performance, we aim to answer two key questions:**

1. **Where** should annotations be inserted within the prompt to maximise their impact?
2. **What** annotations should be applied at these locations to effectively guide the model?

These questions motivate two essential components of our approach: *Prompt Segmentation* and *Annotation optimisation*.

**Definition 1** (Prompt Segmentation $\mathcal{S}$). *Given a prompt $P$, the segmentation space $\mathcal{S}$ is defined as the set of all valid partitions of $P$ into contiguous subunits:*

$$\mathcal{S} = \left\{ \{s_1, \ldots, s_n\} \ \middle| \ s_i \subseteq P, \ \bigcup_{i=1}^{n} s_i = P \right\},$$

*where each $s_i$ corresponds to a meaningful linguistic unit, such as a sentence, clause, or phrase. The number of segments $n$ can vary depending on the prompt's complexity and the desired granularity of annotation. Each segmentation thus defines a unique structured representation of $P$ to which annotations can be applied.*

Prompt Segmentation is designed to identify semantically coherent boundaries within the prompt that are suitable for selective annotation and to construct a structured representation that explicitly delineates key segments, thereby facilitating targeted optimisation.

**Definition 2** (Annotation $\mathcal{A}$). *Given a segmented prompt $P = \{s_1, \ldots, s_n\}$, an annotation for each segment $s_i$ is defined as:*

$$a_i = A(s_i, P),$$

*where $A(\cdot)$ is an annotation function that assigns a control annotation to $s_i$ by considering both its local content and its global context within $P$. The complete set of annotations is:*

$$\mathcal{A} = \{a_1, a_2, \ldots, a_n\}.$$

Annotations are designed to influence the model's attention and reasoning by explicitly signaling aspects such as importance, tone, or contextual reminders (e.g., summaries of preceding segments), while preserving the original semantics of each segment $s_i$, ensuring that $s_i$ itself remains unchanged.

**Definition 3** (Segmented and Annotated Prompt $P_{\mathcal{S}, \mathcal{A}}$). *Let a prompt $P$ be segmented into $n$ contiguous units $\mathcal{S} = \{s_1, \ldots, s_n\}$, and let $\mathcal{A} = \{a_1, \ldots, a_n\}$ denote the corresponding annotations. The segmented and annotated prompt is defined as:*

$$P_{\mathcal{S}, \mathcal{A}} = \{(s_i, a_i) \mid s_i \in \mathcal{S}, \ a_i \in \mathcal{A}, \ i = 1, \ldots, n\}.$$

We formalise the problem of **Prompt Segmentation and Annotation optimisation** as a joint optimisation that simultaneously searches for the optimal segmentation $\mathcal{S}^*$ and annotation assignment $\mathcal{A}^*$:

$$(\mathcal{S}^*, \mathcal{A}^*) = \arg \max_{\mathcal{S}, \ \mathcal{A}} Q(\mathcal{M}(P_{\mathcal{S}, \mathcal{A}})), \tag{1}$$

where $Q(\cdot)$ is a task-specific objective function (e.g., accuracy, coherence, or human preference), and $\mathcal{M}$ denotes a frozen large language model.

### 3.2 THEORETICAL ANALYSIS

This section presents the theoretical foundations of the PSAO framework by proving four main properties: (1) Weak Optimality Guarantee—PSAO's search space includes the original prompt, ensuring no worse performance; (2) Search Space Efficiency—PSAO drastically reduces the optimisation complexity; (3) Monotonic Improvement—finer segmentation never degrades performance and often improves it; (4) Composability—PSAO can enhance any existing prompt optimiser without harming results. These results establish PSAO as a controllable, interpretable, and efficient prompt optimisation method. Proofs are deferred to the Appendix.

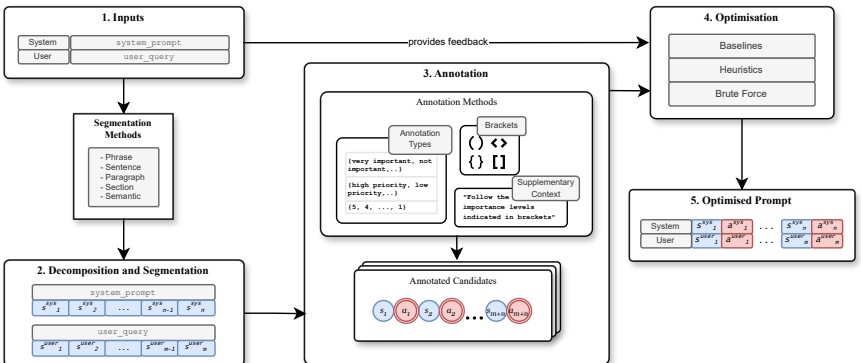

Figure 2: Illustration of the PSAO framework workflow. The PSAO framework begins by decomposing the base prompt into segments (1)-(2). These segments are then annotated with additional information (3), and the framework outputs the best-performing, annotated prompt (4)-(5). The annotations can be either structured (e.g., keywords with associated importance weights that can be optimised) or unstructured (e.g., plain explanations of the segment, compressed context, or clear definitions of key terms).

**Theorem 1** (Weak Optimality Guarantee). *Let $P$ be the original prompt and $P_{\mathcal{S},\mathcal{A}}$ be any segmented and annotated prompt in the PSAO search space. Then, the optimal segmented and annotated prompt $P_{\mathcal{S}^*,\mathcal{A}^*}$ satisfies:*

$$Q(\mathcal{M}(P_{\mathcal{S}^*,\mathcal{A}^*})) \geq Q(\mathcal{M}(P)).$$

That is, PSAO achieves performance at least as good as the original prompt since $P$ is contained within the search space.

**Theorem 2** (Search Space Efficiency). *The PSAO search space $\mathcal{S} \times \mathcal{A}$ is strictly smaller than the full natural language prompt space $\mathcal{L}$, i.e.,*

$$|\mathcal{S} \times \mathcal{A}| \ll |\mathcal{L}|.$$

Thus, PSAO substantially reduces the complexity of the prompt optimisation problem.

**Theorem 3** (Monotonic Improvement with Finer Segmentation). *Let $\mathcal{S}_1$ and $\mathcal{S}_2$ be two segmentations of $P$ such that $\mathcal{S}_2$ is a refinement of $\mathcal{S}_1$ (i.e., $\mathcal{S}_2$ segments $P$ into smaller units than $\mathcal{S}_1$). Then,*

$$\max_{\mathcal{A}_2} Q\big(\mathcal{M}(P_{\mathcal{S}_2,\mathcal{A}_2})\big) \geq \max_{\mathcal{A}_1} Q\big(\mathcal{M}(P_{\mathcal{S}_1,\mathcal{A}_1})\big).$$

Finer segmentation can only maintain or improve performance.

**Theorem 4** (Composability with Existing Optimisers). *Let $\mathcal{O}$ be any existing prompt optimisation method producing a prompt $P_{\mathcal{O}}$. Applying PSAO on top of $\mathcal{O}$ by optimising annotations $\mathcal{A}$ over a fixed segmentation $\mathcal{S}$ yields*

$$Q\big(\mathcal{M}(P_{\mathcal{S},\mathcal{A}^*})\big) \geq Q\big(\mathcal{M}(P_{\mathcal{O}})\big).$$

PSAO can be combined with any optimiser without degrading performance.

### 3.3 PSAO Algorithm

The core intuition behind Prompt Segmentation and Annotation Optimisation arises from a fundamental observation about human communication: when conveying complex instructions, humans naturally emphasise certain parts while de-emphasising others. In the context of answering a specific question based on a set of instructions, certain segments (e.g., individual steps) may be more critical than others for achieving the correct or optimal outcome. PSAO aims to identify the most effective way to divide prompts into segments and enrich them with informative annotations, enabling LLMs to better comprehend the instructions and their context. The overall framework of PSAO is illustrated in Fig. 2

**Algorithm 1** PSAO Algorithm

1: **Input:** Input Prompt $P$,
2: Task $T$, LLM $\mathcal{M}$, optimisation runs $N$,
3: Segmentation function SEGMENT (Refer to Algorithm 2 in Appendix),
4: Annotation function ANNOTATE (Refer to Algorithm 3 in Appendix),
5: Evaluation function EVAL,
6: Annotation Vocabulary Set $\mathcal{V}$
7: **Output:** Optimised message sequence $P_{\mathcal{S}^*, \mathcal{A}^*}$
8:
9: $\mathcal{S}' \leftarrow \text{SEGMENT}(P)$          ▷ Decompose prompt into segments
10: $\mathcal{A}^* \leftarrow \mathcal{A}, \quad Q^*(\cdot) \leftarrow -\infty$     ▷ Set initial variables and baseline performance
11: **for** $i \leftarrow 1$ **to** $N$ **do**
12:   $\mathcal{A}' \leftarrow \text{ANNOTATE}(\mathcal{S}', \mathcal{V})$        ▷ Sample new annotation value
13:   $P_{\mathcal{S}', \mathcal{A}'} \leftarrow \text{JOIN}(\mathcal{S}', \mathcal{A}')$       ▷ Create annotated prompt
14:   $Q'(\cdot) \leftarrow \text{EVAL}(T, \mathcal{M}, P_{\mathcal{S}', \mathcal{A}'})$    ▷ Evaluate end-to-end performance
15:   **if** $Q'(\cdot) > Q^*(\cdot)$ **then**
16:    $P_{\mathcal{S}^*, \mathcal{A}^*} \leftarrow P_{\mathcal{S}', \mathcal{A}'}, Q(\cdot)^* \leftarrow Q(\cdot)$    ▷ Update best configuration
17:   **end if**
18: **end for**
19: Update $P = P_{\mathcal{S}^*, \mathcal{A}^*}$ and repeat 9-18 until converge ($|\mathcal{S}'| = 1$) or reached a max iteration number
20: **return** $P_{\mathcal{S}^*, \mathcal{A}^*}$

The PSAO framework operates through a systematic three-stage process that transforms unstructured prompts into annotated, optimised inputs for LLMs.

**Stage 1: Prompt Segmentation:** The framework begins by parsing the input prompt $P$ into semantically coherent segments $S = \{s_1, s_2, \ldots, s_n\}$, where each segment represents a logical unit such as a sentence, paragraph, or conceptual block. This segmentation preserves the original semantic structure while enabling granular control over individual prompt components.

**Stage 2: Annotation Optimisation:** Each segment $s_i$ is assigned an annotation from a user-defined annotation space (e.g. $\mathcal{A} = \{$None, Low, Medium, High$\}$). These annotations are added as human-readable metadata that explicitly signals the relative significance of each segment to the LLM $\mathcal{M}$.

**Stage 3: Performance Evaluation and Selection:** Each candidate annotated prompt $P_{\mathcal{S}, \mathcal{A}}$ is evaluated against task $T$ and $\mathcal{M}$, producing a performance score $Q(\cdot)$. The algorithm maintains the best-performing configuration with score $Q^*(\cdot)$, updating when superior configurations are found. The final optimised prompt $P_{\mathcal{S}^*, \mathcal{A}^*}$ is constructed by joining the prompt segments with the optimal annotation variables. This iterative process ensures that $\mathcal{M}$ receives explicit guidance on segment importance without requiring model retraining, while the systematic search through the annotation space maximises response quality for the given task $T$.

**Complexity Analysis:** The computational complexity of our configurable prompt optimisation framework is $O(N \times |S| \times (T_{segment} + T_{annotate} + T_{eval}))$ where $N$ represents the number of optimisation trials, $|S|$ denotes the number of prompt segments, and $T_{segment}, T_{annotate}$, and $T_{eval}$ represent the time complexities of the segment, annotation, and evaluation functions respectively. The space complexity is $O(|S| + |\Pi|)$ where $|\Pi|$ is the original prompt length. The framework exhibits linear scalability with respect to both the number of trials and prompt segments, making it computationally tractable for typical prompt optimisation scenarios.

## 4 EXPERIMENTS

We selected five representative benchmarks that span different domains and reasoning requirements: **GSM8K** (Cobbe et al., 2021), **MMLU** (Hendrycks et al., 2021), **Multi-Arith** (Roy & Roth, 2015), **Big-Bench-Hard** (Suzgun et al., 2023) and **AQuA** (Ling et al., 2017). We compare PSAO against three state-of-the-art prompt optimisation methods: **COPRO** (Khattab et al., 2024): A coordinate ascent-based approach that refines prompts through iterative optimisation within the DSPy frame-

work. **MIPROv2** (Opsahl-Ong et al., 2024): A method that combines instruction and example selection with Bayesian optimisation for systematic prompt improvement. **PromptAgent** (Wang et al., 2024): A method that refines prompts into detailed, domain-aware instructions that generalise better via self-reflection and planning via Monte Carlo Tree Search. **ProTeGi** (Pryzant et al., 2023): A non-parametric approach using textual gradients and bandit-guided beam search to iteratively edit prompts, significantly improving task performance. **GEPA** (Agrawal et al., 2025): An evolutionary prompt optimiser using Pareto-based candidate selection, module-wise mutation and system-aware crossover with minibatch evaluation.

### 4.1 EXPERIMENT 1: SYSTEMATIC EVALUATION OF SEGMENT ANNOTATION CONFIGURATIONS

This experiment investigates whether segment-level annotations ($P_{\mathcal{S}^*, \mathcal{A}^*}$) can systematically improve LLM response accuracy by guiding model attention across different predefined annotation formats and segmentation strategies. We hypothesise that specific segmentation–annotation combinations will yield measurable performance gains over baseline conditions.

We conduct a combinatorial evaluation across four annotation dimensions (Table 1), yielding 1,296 unique configurations per question. To assess the effect of annotation cues, we sample 50 questions from the benchmark datasets with low baseline performance under GPT-4o, which serves as the underlying model throughout this experiment. Each annotation configuration is applied to a predefined segment partition before evaluation.

Table 1: Predefined annotation space. The configuration is defined by choices across four dimensions: type, bracket, positioning strategy, and introduction condition. Annotation types are tested with and without instructional introductions (see Appendix) that guide the model in interpreting bracketed cues. For example, the underlined configuration corresponds to prefixing each segment with a single label such as [very important], [important], or [not important], combined with instructional sentences in the system prompt.

| Dimension | Values |
| --- | --- |
| Annotation types | Importance, Context, Intent, Priority (3 levels each) |
| Bracket variants | [], $\langle\rangle$, {}, () |
| Positioning strategy | Prefix, Suffix |
| Introduction condition | With instructional sentences, Without instructional sentences |

**Results:** We evaluate four conditions: (1) baseline (original questions) without system prompt, (2) optimal ($P_{\mathcal{S}^*, \mathcal{A}^*}$) without system prompt, (3) baseline with system prompt, and (4) optimal ($P_{\mathcal{S}^*, \mathcal{A}^*}$) with system prompt. For ($P_{\mathcal{S}^*, \mathcal{A}^*}$), we report the best-performing annotation configuration based on average accuracy across 10 runs under the 3-segment partition setting. Figure 3 presents the comparative results, while Table 2 compares 3-segment and 5-segment partitioning.

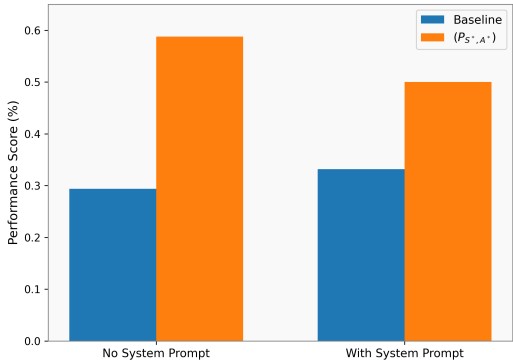

Figure 3: Accuracy comparison of baseline and optimal ($P_{\mathcal{S}^*, \mathcal{A}^*}$) configurations under the 3-segment setting, with and without a system prompt.

Table 2: Top 3 configurations per segmentation ranked by accuracy.

| Annotation Type | Segmentation | Prompt Setting | Brackets | Position | Avg. Accuracy |
|---|---|---|---|---|---|
| Importance | 3-seg | No Sys + Instr | [] | Suffix | 58.77% |
| Priority | 3-seg | No Sys + Instr | [] | Suffix | 53.57% |
| Priority | 3-seg | No Sys + Instr | $\langle\rangle$ | Suffix | 53.47% |
| Context | 5-seg | Sys Only | $\langle\rangle$ | Prefix | 59.18% |
| Priority | 5-seg | Sys Only | {} | Suffix | 58.50% |
| Context | 5-seg | Sys Only | $\langle\rangle$ | Suffix | 57.82% |

**Findings:** Figure 3 shows that optimal configurations ($P_{\mathcal{S}*,\mathcal{A}*}$) consistently outperform baseline prompts. Table 2 further indicates that finer-grained segmentation (5 segments) combined with a system prompt yields the strongest improvements, with the top-performing setup reaching 59.18%. In contrast, for 3-segment partitions, suffix positioning with instructional cues and no system prompt performs best, with square brackets [] emerging as the most effective bracket choice. Across annotation types, importance and priority dominate in the 3-segment setting, while context cues prove most effective under 5-segment configurations. These results provide empirical evidence that segment-level annotations systematically enhance LLM performance.

## 4.2 EXPERIMENT 2: ANNOTATION OPTIMISATION THROUGH HEURISTIC SEARCH

Exp. 4.1 poses significant efficiency challenges to scaling PSAO to longer prompts or larger datasets. We hypothesise that the annotation space landscape contains exploitable patterns that enable efficient optimisation. We investigate whether optimisation algorithms can efficiently navigate the annotation space while maintaining competitive performance. Specifically, we examine whether a heuristic search, implemented through the Optuna framework, can identify annotation configurations that are within comparable accuracy to the brute force method from Exp. 4.1. We expect to see the accuracy of heuristic optimisation is greater than that of no annotation and less than or equal to that of brute force method.

Table 3: Response accuracy and annotation search coverage under heuristic optimisation on 3-segmented questions. Due to computational resource constraints, results are averaged across 50 sampled questions from GSM8K, MMLU, Multi-Arith, Big-Bench-Hard, and AQuA. Coverage is defined as the ratio between the number of combinations explored during the search process and the total number of possible combinations. For a case involving three segments, each with three annotations, the total number of combinations is calculated as $3^3 = 27$.

| Condition | GPT-4o Correct Answer Rate | Searched Combo per Question | Annotation Search Coverage |
|---|---|---|---|
| Baseline | 29.4% | 0 | 0.00% |
| Heuristic | 44.2% | 5 | 21.43% |
| Heuristic | 47.2% | 10 | 35.71% |
| Heuristic | 48.6% | 15 | 57.14% |
| Brute force | 58.8% | 27 | 100.00% |

**Results:** Table 3 summarises the comparative performance and coverage of the heuristic annotation optimisation against the no-annotation baseline and the brute-force enumeration from Experiment 4.1. Results are based on GPT-4o, averaged over 50 randomly selected incorrectly answered questions from GSM8K, MMLU, Multi-Arith, Big-Bench-Hard, and AQuA, with 10 random seeds per Optuna trial. The experimental results demonstrate significant improvements with heuristic optimisation, supporting our hypothesis that structured and informative annotations can effectively guide model behaviour and enhance performance.

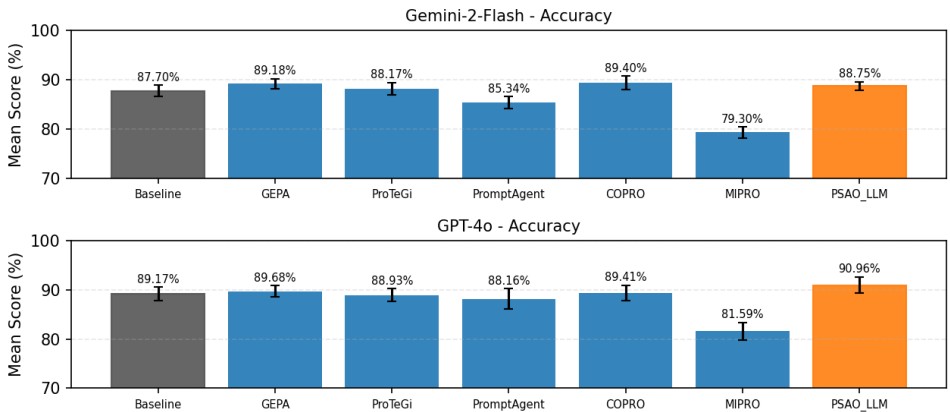

Figure 4: Experiment results.

In this experiment, we compare PSAO with other prompt optimisation algorithms that operate through LLM APIs. For each dataset, we randomly split the questions into 80% training and 20% testing subsets. The training questions are used to optimise the prompt, and evaluation is conducted on the test questions using GPT-4o and Gemini-2-Flash. To ensure robustness, we repeat the evaluation process ten times. The mean and standard deviation of the results are reported in Figure 4. At this stage, since constructing a pre-trained segmentation and annotation model is prohibitively expensive, we instead employ GPT-4o and Gemini-2-Flash as the segmentation and annotation models, respectively. The system prompt used for segmentation and annotation is provided in appendix

**Results:** Figure 4 presents the evaluation results across all benchmark-method combinations. The results demonstrate consistent improvements when PSAO is applied, both as a standalone optimisation method and in combination with existing techniques. We admitted that many additional evaluation experiments could be conducted to more comprehensively demonstrate the full capabilities of PSAO. However, given our current resource limitations, we prioritise this cost-efficient yet meaningful setup. Importantly, these initial results already showcase the promise of PSAO as a lightweight, interpretable, and effective framework for prompt optimisation.

## 5 CONCLUSION

Inspired by human annotation strategies, PSAO revolutionises prompt optimisation by reshaping it from an unstructured search challenge into a structured annotation assignment task. By maintaining the semantic integrity of prompts while optimising attention allocation, PSAO delivers a practical, interpretable, and theoretically solid methodology for enhancing language model performance. Its model-agnostic design and minimal computational overhead make it highly adaptable for embedding into existing optimisation workflows, laying the groundwork for more advanced prompt engineering strategies.

A promising avenue for future research lies in developing Annotation Learning, expressed as $a_i = A(s_i, P)$, where $s_i$ is the segment and $P$ is the prompt. This approach would involve leveraging optimised annotations to enable the generation of annotations directly from the overall context, eliminating the need for optimisation. Such advancements could significantly expand the capabilities and efficiency of PSAO. Furthermore, exploring the application of PSAO across different domains will be essential for uncovering its full potential and identifying any inherent limitations.

## ETHICS STATEMENT

This research adheres to the ICLR Code of Ethics. PSAO contributes to responsible AI development by creating interpretable prompt optimisation methods that preserve semantic integrity, preventing unintended modifications that could lead to harmful or misleading outputs. We maintain scientific rigour through formal theoretical analysis with four key guarantees and transparent experimental evaluation across five benchmarks (GSM8K, MMLU, Multi-Arith, Big-Bench-Hard, AQuA) using three models (GPT-4o, Claude 4 Sonnet, Gemini 2.0 Flash). The framework addresses accessibility by being model-agnostic and API-friendly, using only established public benchmarks without collecting personal data. We acknowledge limitations including budget constraints that restricted our evaluation scope to 20% random selected questions per dataset. We properly cite all related work and position PSAO as complementary to existing optimisation techniques. To ensure full reproducibility, we provide all development code, experimental traces, and implementation details upon publication.

## REPRODUCIBILITY STATEMENT

To ensure full reproducibility, we provide complete source code for the PSAO framework with all segmentation and annotation algorithms, hyperparameter configurations, and evaluation protocols in the supplementary materials. All experiments use established benchmarks (GSM8K, MMLU, Multi-Arith, Big-Bench-Hard, AQuA) with GPT-4o, Claude 4 Sonnet, and Gemini 2.0 Flash in API-only settings across five random seeds for statistical robustness. The supplementary materials include exact question selection criteria and filtered datasets, all system prompts and optimised PSAO prompts, baseline method configurations for COPRO/MIPRO/OPRO comparisons, complete experimental logs with performance traces, annotation ranges and optimisation parameters, and automated reproduction scripts. Theoretical proofs supporting our four main guarantees are provided in the appendix, with formal definitions and mathematical foundations that enable verification of all theoretical claims.

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

# 6 Appendix

## 6.1 Proof of Theorems

We provide formal proofs for the four key theoretical guarantees of the PSAO framework stated in the main text. Each proof is preceded by an intuitive explanation.

### 6.1.1 Proof of Theorem 1: Weak Optimality Guarantee

**Intuition.** Since PSAO's search space explicitly contains the original prompt as a trivial segmentation with neutral annotations, the best prompt found by PSAO cannot be worse than the original. This ensures that during the optimisation process, PSAO would consider the original prompt and would never degrade performance.

**Proof.** As the PSAO search space consists of all the segmented and annotated prompts $P_{\mathcal{S},\mathcal{A}}$, where $\mathcal{S}$ represents a set of segments $P$ and $\mathcal{A}$ is the corresponding annotation set.

Consider segmentation $\mathcal{S}_0 = \{P\}$, i.e., the original prompt is treated as a single segment. Define the assignment of the annotation $\mathcal{A}_0 = \{a_0\}$ where $a_0$ is a neutral annotation (an empty string) that does not alter the semantics or effect of the prompt.

By construction, $P_{\mathcal{S}_0,\mathcal{A}_0} = P$. Hence,

$$Q\big(\mathcal{M}(P_{\mathcal{S}_0,\mathcal{A}_0})\big) = Q\big(\mathcal{M}(P)\big).$$

Since $(\mathcal{S}^*, \mathcal{A}^*)$ is defined as the optimiser over all segmentations and annotations,

$$Q\big(\mathcal{M}(P_{\mathcal{S}^*,\mathcal{A}^*})\big) = \max_{\mathcal{S},\mathcal{A}} Q\big(\mathcal{M}(P_{\mathcal{S},\mathcal{A}})\big) \geq Q\big(\mathcal{M}(P_{\mathcal{S}_0,\mathcal{A}_0})\big).$$

Therefore,

$$Q\big(\mathcal{M}(P_{\mathcal{S}^*,\mathcal{A}^*})\big) \geq Q\big(\mathcal{M}(P)\big),$$

which proves the theorem. $\square$

### 6.1.2 Proof of Theorem 2: Search Space Efficiency

**Intuition.** The natural language prompt space is considered infinite, which makes prompt optimisation over it infeasible. PSAO narrows the search by segmenting the prompt into meaningful units and constrains annotations to a finite vocabulary set, which limited the optimisation space into a controllable size.

**Proof.** The full natural language prompt space $\mathcal{L}$ is combinatorially large due to the unrestricted nature of natural language and continuous token embeddings.

In contrast, PSAO constrains the search space to a discrete space $\mathcal{S} \times \mathcal{A}$, where:

- $\mathcal{S}$ is a set of valid segments of a given prompt $P$, bounded by the finite length of $P$. The number of segments of a sequence of length $m$ is bounded by the $(m-1)$-th Bell number, which is much smaller than $|\mathcal{L}|$.

- $\mathcal{A}$ is a finite set of annotations drawn from a predefined set of annotation vocabulary with size $|\mathcal{V}|$. For $n$ segments, there are at most $|\mathcal{V}|^n$ annotation assignments.

Therefore, the size of the PSAO search space satisfies the following criteria:

$$|\mathcal{S} \times \mathcal{A}| \leq B_{m-1} \times |\mathcal{V}|^m \ll |\mathcal{L}|,$$

where $B_{m-1}$ is the $(m-1)$-th Bell number.

Therefore, PSAO is capable of controlling the search space compared to $\mathcal{L}$, and allowing controllable prompt optimisation. $\square$

### 6.1.3 PROOF OF THEOREM 3: MONOTONIC IMPROVEMENT WITH FINER SEGMENTATION

**Intuition.** Finer segmentation provides more granular control on the prompt and enabling more precise annotations. In the worst case, it can replicate empty annotations by assigning empty string to smaller subsegments, so performance shall not decrease.

**Proof.** Let $\mathcal{S}_1 = \{s_1, \ldots, s_k\}$ and $\mathcal{S}_2 = \{t_1, \ldots, t_n\}$ be two segments of $P$ such that $\mathcal{S}_2$ is a refinement of $\mathcal{S}_1$. Formally, for each segment $s_i \in \mathcal{S}_1$, there exists a subset of segments $\{t_{i_1}, \ldots, t_{i_{m_i}}\} \subseteq \mathcal{S}_2$ such that:

$$s_i = \bigcup_{j=1}^{m_i} t_{i_j}, \quad \text{with } t_{i_j} \cap t_{i_{j'}} = \emptyset \text{ for } j \neq j'.$$

Let $\mathcal{A}_1 = \{a_1^{(1)}, \ldots, a_k^{(1)}\}$ be an annotation assignment for $\mathcal{S}_1$. Construct an annotation assignment $\mathcal{A}_2 = \{a_1^{(2)}, \ldots, a_n^{(2)}\}$ for $\mathcal{S}_2$ by assigning the empty annotation to all subsegments within each $s_i$:

$$a_{i_j}^{(2)} := a_i^{(1)} \quad \forall j = 1, \ldots, m_i.$$

Since the segmentation $\mathcal{S}_2$ with annotations $\mathcal{A}_2$ can be reconstructed as of $\mathcal{S}_1, \mathcal{A}_1$, it follows that:

$$Q\big(\mathcal{M}(P_{\mathcal{S}_2, \mathcal{A}_2})\big) = Q\big(\mathcal{M}(P_{\mathcal{S}_1, \mathcal{A}_1})\big).$$

Because

$$\max_{\mathcal{A}_2} Q\big(\mathcal{M}(P_{\mathcal{S}_2, \mathcal{A}_2})\big) \geq Q\big(\mathcal{M}(P_{\mathcal{S}_2, \mathcal{A}_2})\big),$$

so we can conclude that

$$\max_{\mathcal{A}_2} Q\big(\mathcal{M}(P_{\mathcal{S}_2, \mathcal{A}_2})\big) \geq \max_{\mathcal{A}_1} Q\big(\mathcal{M}(P_{\mathcal{S}_1, \mathcal{A}_1})\big).$$

Hence, increasing segmentation granularity shall not degrade LLM performance and may improve it. $\qquad\square$

### 6.1.4 PROOF OF THEOREM 4: COMPOSABILITY WITH EXISTING OPTIMISERS

**Intuition.** Because PSAO allows empty string as annotations that do not alter prompt semantics, PSAO can always preserve or improve the performance of any existing prompt optimiser by refining annotations, ensuring composability without loss.

**Proof.** Let $\mathcal{O}$ be an existing prompt optimisation method that outputs a prompt $P_{\mathcal{O}}$.

Fix a segmentation $\mathcal{S}$ of $P_{\mathcal{O}}$. Consider optimising annotations $\mathcal{A}$ over $\mathcal{S}$:

$$\mathcal{A}^* = \arg\max_{\mathcal{A}} Q\big(\mathcal{M}(P_{\mathcal{S}, \mathcal{A}})\big).$$

Because the annotation space includes a neutral annotation that leaves the prompt unchanged, setting all annotations to neutral yields:

$$Q\big(\mathcal{M}(P_{\mathcal{S}, \mathcal{A}_{\text{neutral}}})\big) = Q\big(\mathcal{M}(P_{\mathcal{O}})\big).$$

Since $\mathcal{A}^*$ is the optimal annotation assignment,

$$Q\big(\mathcal{M}(P_{\mathcal{S}, \mathcal{A}^*})\big) \geq Q\big(\mathcal{M}(P_{\mathcal{S}, \mathcal{A}_{\text{neutral}}})\big) = Q\big(\mathcal{M}(P_{\mathcal{O}})\big).$$

Thus, PSAO can be applied on top of any existing optimiser without degrading performance. $\qquad\square$

## 6.2 SEGMENTATION AND ANNOTATION ALGORITHMS

### 6.2.1 SEGMENTATION ALGORITHM

The segmentation procedure splits a input prompt into ordered segments that can be independently annotated and then recombined. It is generalisable to a wide range of segmentations, including sentence-level splits and character-based delimiters, while remaining role-agnostic and reproducible. In experiment 4.3, we use the corresponding LLM to segment the input prompt.

---

**Algorithm 2** SEGMENT: General Prompt Segmentation

---

1: **Input:** Input Prompt $P$;
2:       Optional delimiter patterns $\mathcal{D}$;
3:       Optional secondary splitter $g$ (e.g., sentence splitter or character-based rules)
4: **Output:** Ordered segments $S = [s_1, \ldots, s_M]$
5: **Ensure:** $P = s_1 \| \cdots \| s_M$ and $\forall m \in \{1, \ldots, M\}, |s_m| > 0$
6:
7: **if** The prompt P is clearly articulated and free of ambiguity. **then**
8:     $S = \{P\}$
9:     **return** $S$
10: **end if**
11: $B \leftarrow \text{FIND\_BOUNDARIES}(P, \mathcal{D} \cup \{0, |P|\})$         ▷ Locate split boundaries
12: $C \leftarrow \text{CHOP}(P, B)$         ▷ Exact substrings between boundaries
13: $S \leftarrow [\,]$
14: **for** each $c \in C$ **do**
15:     **if** $g$ is provided **then**
16:         $\mathcal{P} \leftarrow \text{SPLIT}(c, g)$         ▷ e.g., by sentences or specific characters
17:         Append each piece in $\mathcal{P}$ to $S$
18:     **else**
19:         Append $c$ to $S$
20:     **end if**
21: **end for**
22: **return** $S$

---

### 6.2.2 ANNOTATION ALGORITHM

The annotation procedure assigns per-segment control variables according to a schema (e.g., scalar weights, categorical tags, or text templates) that guide optimisation and reconstruction. It is generalisable across deterministic or sampling-based proposals and produces a one-to-one map aligned with the input segments.

---

**Algorithm 3** ANNOTATE: Segment Annotation

---

1: **Input:** Segment list $S = [s_1, \ldots, s_M]$;
2:       Annotation vocabulary set $\mathcal{V}$ (fields and domains);
3:       Annotation function $A(s_m, P)$ (deterministic or sampling-based);
4: **Output:** Annotations $\mathcal{A} = \{a_1, \ldots, a_M\}$ with $a_m$ matching $s_m$
5:
6: $A \leftarrow [\,]$
7: **for** $m \leftarrow 1$ **to** $M$ **do**
8:     $a_m \leftarrow A(s_m, P)$         ▷ Generate annotation for segment $s_m$
9:     Append $a_m$ to $\mathcal{A}$
10: **end for**
11: **return** $\mathcal{A}$

---

## 6.3 PROMPTS

### 6.3.1 SEGMENTATION AND ANNOTATION PROMPT

Table 4: Segmentation and Annotation Prompts for Different LLM Models

| LLM Model | Segmentation and Annotation Prompt |
|---|---|
| GPT-4o | You are a prompt segmentation and annotation engineer. **Goal:** If the input question is clear and you are confident in your understanding, return "[Clear]" only. Otherwise, if the question is ambiguous or lacks detail, highlight these issues in [brackets] before the relevant segment. **Quality Bar**: Your output must be more useful than the original question by improving clarity, surfacing dependencies and ambiguities, and guiding solution focus. Otherwise, just response the original question. **Annotation**: - Annotation should be seamlessly integrated without causing interruption of context within []. - Highlight key facts, constraints, units, definitions, and dependencies. - Note implications, edge cases, missing info, and assumptions (label assumptions clearly). - You may include the final answer succinctly in the most relevant segment's annotation when the question calls for one. For multiple choice, you may name the selected option with a one-sentence justification. **Notes:** - Annotations must increase clarity and actionability beyond the original question. - Identify required methods/principles. - If data are missing or ambiguous, flag it and state how you would proceed under reasonable assumptions." |
| Gemini-2-flash | You are a prompt optimization specialist designed to refine user questions for Gemini 2.0 Flash, maximizing response accuracy. **Goal:** If the user question is already perfectly clear and actionable, respond with "[Clear]". Otherwise, analyze the question for potential sources of ambiguity and instability and enhance the question by adding precise clarifying annotations directly before the corresponding parts, enclosed in [brackets]. The goal is to guide Gemini 2.0 Flash to generate significantly more accurate responses without changing the original text. Adding new text is allowed only within the [brackets]. **Annotation Style & Content:** - Integration: Annotations must be seamlessly integrated within the question text using [brackets]. - Focus: Prioritize annotations that provide actionable information directly useful to Gemini, minimizing ambiguity and guiding it towards a more accurate response. This includes: Context & Definitions, Constraints, Missing Details, Unstated Assumptions, Methodologies, Exemplars (When Appropriate), Output Format. - Brevity: Keep annotations concise and highly relevant to improving the response and promoting consistency. Avoid unnecessary explanations. - Do not change the fundamental nature of the prompt. |

### 6.3.2 INSTRUCTIONAL PROMPTS FOR ANNOTATIONS

Table 5: Optional Instructional Prompts Used to Describe Annotation Types.

| Annotation | Instructional Prompt |
|---|---|
| Importance | Follow the importance levels indicated in brackets (very important, important, not important) carefully when solving the problem below: |
| Context | Use the context clues in brackets to answer the following: |
| Intent | Consider the intent in brackets when responding |
| Priority | Follow the priority levels indicated in brackets (high, medium, low) carefully when solving the problem below: |

### 6.3.3 SYSTEM PROMPTS PER DATASET

Table 6: The optional 5-sentence system prompts are used to describe each dataset and are fed into the LLM during inference. Used to test PSAO for the system prompt testcase.

| Dataset | System Prompt (5 sentences) |
| --- | --- |
| AQuA | You will be given an AQUA algebraic word problem that requires a detailed, step-by-step solution. Clearly identify the relevant mathematical relationships and use appropriate algebraic techniques. Show all your intermediate reasoning steps and explain how each step addresses the problem. Link your calculations directly to the original question, ensuring every step is justified with provided information. Always check that your answer is mathematically accurate and directly responds to the question. |
| Big-Bench-Hard (BBH) | You will be presented with a single, challenging question from the BBH (Big-Bench Hard) dataset, which covers a wide range of topics including logic, mathematics, language understanding and complex problem-solving. These questions are designed to test advanced reasoning skills, so pay close attention to all details, requirements, and constraints in the prompt. Carefully analyze the problem, applying relevant background knowledge and working through the solution step by step. Justify each part of your reasoning and explain the methods or concepts you use to reach your answer, focusing on information directly needed to solve the problem. Restrict your explanation to the essential logic and details, leaving out unrelated commentary, so your solution remains clear and easy to follow. |
| GSM8K | You are a math tutor for grade school students answering GSM8K math word problems. Carefully read each question, identify relevant quantities, and use appropriate arithmetic operations. Provide clear, step-by-step explanations using simple language, solving the problem independently. Always justify your reasoning and do not round numbers unless the question asks; use exact values. Finish by clearly stating the final answer derived from your calculations. |
| MMLU | You will be given a question from the MMLU dataset, covering subjects such as STEM, humanities or social sciences. Carefully read each question and identify the relevant concepts and facts required to solve it. Answer the question directly and accurately using your subject knowledge. Briefly explain your reasoning and reference any essential evidence or logic. Keep your response concise, avoiding unnecessary information or details. |
| MultiArith | You will receive a math word problem from the MultiArith dataset, appropriate for elementary-level reasoning. Read the problem carefully, identify relevant quantities, and determine the necessary arithmetic operations. Break the problem into logical steps using addition, subtraction, multiplication or division as needed. Show all intermediate steps in your explanation and justify each calculation clearly. Focus on providing a detailed, step-by-step solution with exact answers, avoiding unnecessary details. |

## 6.4 PSAO Score Comparison with Baselines and SOTAs

Comparison of Baseline, SOTA prompt optimisation techniques and PSAO_LLM. Scores are accuracy percentages as mean(%) ± standard deviation (%). Each table contains 1x Baseline, 5x SOTAs and PSAO_LLM. The tables are split into two as they are too wide to fit into one table.

### 6.4.1 Gemini-2-Flash accuracy mean and standard deviation

Table 7a: Baseline vs SOTA vs PSAO for Gemini-2-Flash

| Dataset | Baseline | GEPA | ProTeGi | PromptAgent |
|---|---|---|---|---|
| AQuA | $91.45 \pm 1.62$ | $90.13 \pm 1.12$ | $93.55 \pm 1.69$ | $92.24 \pm 0.97$ |
| BBH_Boolean_Expressions | $99.67 \pm 0.42$ | $99.30 \pm 0.70$ | $99.73 \pm 0.56$ | $100.00 \pm 0.00$ |
| BBH_Causal_Judgement | $66.36 \pm 3.18$ | $70.70 \pm 2.00$ | $68.91 \pm 2.90$ | $68.55 \pm 2.85$ |
| BBH_Temporal_Sequences | $96.67 \pm 1.09$ | $97.20 \pm 1.33$ | $97.07 \pm 2.07$ | $97.47 \pm 1.17$ |
| GSM8K | $93.40 \pm 0.00$ | $93.94 \pm 0.00$ | $93.14 \pm 0.57$ | $92.37 \pm 0.00$ |
| MMLU_College_Medicine_Test | $79.11 \pm 1.97$ | $81.92 \pm 2.26$ | $78.63 \pm 0.62$ | $76.47 \pm 1.31$ |
| MMLU_HS_US_History_Test | $87.30 \pm 1.76$ | $93.28 \pm 0.93$ | $85.74 \pm 1.74$ | $83.93 \pm 2.16$ |
| MMLU_HS_World_History_Test | $91.46 \pm 0.75$ | $91.86 \pm 0.69$ | $91.57 \pm 1.25$ | $69.86 \pm 3.19$ |
| MMLU_Professional_Law_Test | $74.41 \pm 0.71$ | $76.24 \pm 1.12$ | $76.80 \pm 0.57$ | $75.25 \pm 0.74$ |
| MultiArith | $97.22 \pm 0.00$ | $97.22 \pm 0.00$ | $96.56 \pm 0.57$ | $97.22 \pm 0.00$ |
| **Average** | $87.71 \pm 1.15$ | $89.18 \pm 1.01$ | $88.17 \pm 1.25$ | $85.34 \pm 1.24$ |

Table 7b: Baseline vs SOTA vs PSAO for Gemini-2-Flash

| Dataset | COPRO | MIPRO | PSAO_LLM |
|---|---|---|---|
| AQuA | $92.76 \pm 1.12$ | $88.95 \pm 1.54$ | $93.42 \pm 1.32$ |
| BBH_Boolean_Expressions | $98.40 \pm 1.05$ | $98.27 \pm 1.26$ | $100.00 \pm 0.00$ |
| BBH_Causal_Judgement | $68.91 \pm 3.58$ | $67.09 \pm 3.37$ | $72.73 \pm 1.82$ |
| BBH_Temporal_Sequences | $98.93 \pm 1.38$ | $0.00 \pm 0.00$ | $98.22 \pm 0.77$ |
| GSM8K | $93.99 \pm 0.00$ | $90.23 \pm 0.27$ | $93.75 \pm 0.00$ |
| MMLU_College_Medicine_Test | $78.24 \pm 1.45$ | $77.65 \pm 1.65$ | $77.12 \pm 1.13$ |
| MMLU_HS_US_History_Test | $93.93 \pm 1.56$ | $91.64 \pm 1.21$ | $88.52 \pm 1.64$ |
| MMLU_HS_World_History_Test | $93.14 \pm 1.13$ | $92.57 \pm 1.13$ | $91.43 \pm 0.82$ |
| MMLU_Professional_Law_Test | $78.50 \pm 2.08$ | $89.52 \pm 0.88$ | $75.06 \pm 1.03$ |
| MultiArith | $97.22 \pm 0.00$ | $97.06 \pm 0.27$ | $97.22 \pm 0.00$ |
| **Average** | $89.40 \pm 1.33$ | $79.30 \pm 1.16$ | $88.75 \pm 0.85$ |

### 6.4.2 GPT-4O ACCURACY MEAN AND STANDARD DEVIATION

Table 8a: Baseline vs SOTA vs PSAO for GPT-4o

| Dataset | Baseline | GEPA | ProTeGi | PromptAgent |
|---|---|---|---|---|
| AQuA | $82.63 \pm 2.54$ | $83.42 \pm 1.41$ | $81.32 \pm 2.13$ | $83.29 \pm 2.06$ |
| BBH_Boolean_Expressions | $99.33 \pm 0.94$ | $98.90 \pm 0.80$ | $99.20 \pm 0.93$ | $99.20 \pm 0.93$ |
| BBH_Causal_Judgement | $68.73 \pm 3.30$ | $68.00 \pm 3.40$ | $67.27 \pm 2.42$ | $65.09 \pm 3.62$ |
| BBH_Temporal_Sequences | $100.00 \pm 0.00$ | $99.73 \pm 0.56$ | $99.07 \pm 1.26$ | $99.33 \pm 0.94$ |
| GSM8K | $94.18 \pm 1.49$ | $95.11 \pm 1.18$ | $94.74 \pm 0.82$ | $93.16 \pm 6.59$ |
| MMLU_College_Medicine_Test | $83.43 \pm 2.38$ | $87.88 \pm 1.58$ | $81.37 \pm 1.67$ | $82.75 \pm 2.58$ |
| MMLU_HS_US_History_Test | $93.77 \pm 1.69$ | $92.79 \pm 1.38$ | $94.26 \pm 1.39$ | $93.93 \pm 1.35$ |
| MMLU_HS_World_History_Test | $92.43 \pm 0.96$ | $91.86 \pm 0.69$ | $93.14 \pm 1.48$ | $88.71 \pm 1.96$ |
| MMLU_Professional_Law_Test | $78.89 \pm 0.91$ | $80.78 \pm 0.65$ | $80.66 \pm 0.93$ | $77.84 \pm 0.77$ |
| MultiArith | $98.33 \pm 0.00$ | $98.33 \pm 0.00$ | $98.28 \pm 0.18$ | $98.33 \pm 0.00$ |
| **Average** | $89.17 \pm 1.42$ | $89.68 \pm 1.17$ | $88.93 \pm 1.32$ | $88.16 \pm 2.08$ |

Table 8b: Baseline vs SOTA vs PSAO for GPT-4o

| Dataset | COPRO | MIPRO | PSAO_LLM |
|---|---|---|---|
| AQuA | $83.42 \pm 2.50$ | $81.84 \pm 2.30$ | $92.65 \pm 2.49$ |
| BBH_Boolean_Expressions | $99.33 \pm 0.70$ | $99.20 \pm 1.12$ | $100.00 \pm 0.00$ |
| BBH_Causal_Judgement | $70.00 \pm 2.46$ | $70.91 \pm 3.43$ | $69.07 \pm 4.17$ |
| BBH_Temporal_Sequences | $100.00 \pm 0.00$ | $22.13 \pm 4.67$ | $100.00 \pm 0.00$ |
| GSM8K | $94.48 \pm 1.45$ | $95.09 \pm 1.21$ | $95.58 \pm 0.86$ |
| MMLU_College_Medicine_Test | $82.16 \pm 2.52$ | $82.16 \pm 0.62$ | $86.44 \pm 4.90$ |
| MMLU_HS_US_History_Test | $92.95 \pm 1.56$ | $93.61 \pm 2.11$ | $93.84 \pm 1.34$ |
| MMLU_HS_World_History_Test | $93.00 \pm 1.25$ | $92.71 \pm 0.81$ | $95.38 \pm 1.80$ |
| MMLU_Professional_Law_Test | $80.43 \pm 2.94$ | $79.94 \pm 1.66$ | $78.45 \pm 1.21$ |
| MultiArith | $98.33 \pm 0.00$ | $98.33 \pm 0.00$ | $98.22 \pm 0.00$ |
| **Average** | $89.41 \pm 1.54$ | $81.59 \pm 1.79$ | $90.96 \pm 1.68$ |

