# OpenReview forum: "Prompt Segmentation and Annotation Optimisation: Controlling LLM Behaviour via Optimised Segment-Level Annotations"
_ICLR.cc/2026/Conference — ICLR 2026 Conference Withdrawn Submission_

### Official Review · Reviewer_mcbE · 2025-10-29

**Soundness:** 2
**Presentation:** 2
**Contribution:** 2
**Rating:** 2
**Confidence:** 4

**Summary:**

This paper introduces Prompt Segmentation and Annotation Optimization, a framework that aims to improve LLM performance by segmenting a prompt into interpretable sub-units and adding lightweight, human-readable annotations. These annotations are meant to guide model attention and reasoning without altering the original semantics.

**Strengths:**

1. The related work section is broad and accurately situates PSAO among recent prompt optimization and meta-prompting methods.
2. The paper is well-written is easy to get understood.

**Weaknesses:**

1. The experiments are small-scale and largely illustrative. Results are based on 50 sampled questions per dataset, which is insufficient for statistical rigor. Only accuracy is reported; no metrics for computational cost, generalization, or human interpretability are shown. Only GPT-4o based results are reported, without showing the generalizability of the PSAO on other model architectures.
2. The experimental section omits several crucial baselines that are conceptually and functionally close to PSAO, such as Reflexion, Self-Correction, and Self-Consistency. These approaches already provide structured prompt-level reasoning improvements or iterative self-refinement, often with fewer model calls. Without direct comparison, it is unclear whether PSAO offers any real advantage over these simpler and widely known methods. Besides accuracy, how’s the efficiency of PSAO compared with these baselines?  The pipeline would add multiple inference steps per input, which would substantially increase computational cost compared to methods that typically involve a few inference passes for self-revision.
3. The assumption that annotations like important actually modulate LLM attention lacks empirical evidence. No ablation study or analysis demonstrates how or why annotations change output behaviour.
4. The theoretical analysis restates that refining segmentation cannot worsen performance, but could not show that the PSAO can essentially benefit the performance.

**Questions:**

See above

---

### Official Review · Reviewer_Nnve · 2025-10-31

**Soundness:** 2
**Presentation:** 4
**Contribution:** 2
**Rating:** 2
**Confidence:** 4

**Summary:**

This paper introduces a new prompt optimisation framework called prompt segmentation and annotation optimisation (PSAO), which decomposes a prompt into segments that are then annotated by an LLM, e.g. according to how important that segment is to the overall prompt. These segments and annotations are concatenated, guiding an LLM to allocate attention more effectively. The authors show that PSAO does not worsen performance compared to baseline and provide experiments on a diverse set of benchmarks, comparing to other state-of-the-art prompt optimisation methods.

**Strengths:**

- Clear motivation; prompt optimisation is an important area; limitations are known
- High quality of writing, very clear style and easy to follow
- Formulas and algorithms contribute to clarity
- Authors compare their method to a number of state-of-the art prompt optimisation methods
- The method is easy to apply, interpretable and practical

**Weaknesses:**

- Largest weakness: the contribution seems to be quite small and lacking in novelty, see e.g. https://arxiv.org/abs/2412.03556
- Best-of-N prompting literature is not mentioned
- The theoretical claims are presented very clearly, but quite weak.
- There should be more baselines: e.g. segmenting the text randomly, adding random labels, or just randomly varying a prompt
- The result seems to suggest that just varying the prompt will lead to at least some configurations that outperform the original prompt, but that seems like a very weak claim. What is the average across all configurations?
- The brute-force style of the algorithm makes the method quite computationally inefficient
- The choice of prompt configuration is based on the evaluated output; in general, we might not have access to that. There seems to be no generalization.


Nitpicks:
- Figure 1 is too small; text is barely readable. I would suggest keeping the figure more schematic/abstract and replacing the detailed prompt.
- The dimensions varied in the experiments seem arbitrary. Why include bracket variants?
- Using more models would have strengthened the claims
- in Table 3, how do the three "Heuristic" Conditions differ?
- What significance level is used for the significance testing
- Missing reference to appendix in line 459

**Questions:**

- In line 339 "yielding 1,296 unique configurations" - how is that related to Table 1? That only seems to account for 5 x 4 x 2 x 2 combinations.
- In line 455 "The training questions are used to optimize the prompt"--> since the segmentation and labels are unique for each prompt, how is the training data used for prompt optimization?

---

### Official Review · Reviewer_wLP6 · 2025-10-31

**Soundness:** 1
**Presentation:** 2
**Contribution:** 2
**Rating:** 2
**Confidence:** 4

**Summary:**

The paper proposes prompt segmentation and annotation optimization (PSAO) for automatic prompt optimization. The method involves two major steps: (1) segmenting an original prompt into segments; (2) annotate each segment as not important, important or very important. Three sets of experiments were performed using PSAO: (1) a brute-force search over different choices for the segmentation (3 segments or 5 segments) or annotation (using 1/2/3 or not important/important/very important). (2) running heuristic searches to show that this also helps but underperforms brute-force search. (3) comparison with other automatic prompt optimization methods.

**Strengths:**

* PSAO is a easy-to-implement and intuitive method for prompt optimization. It turns unstructured prompt editing into more structured edits so the method may benefit from optimization methods like heuristic search.

**Weaknesses:**

* Experiment details are missing. See questions below.
* Unclear benefits of using PSAO. The method does not show significant advantages over baselines according to Figure 4.
* Claims without sufficient support. In the abstract: "[PSAO] can be seamlessly integrated with existing prompt optimisation methods." However I didn't find the corresponding experiments.

**Questions:**

* Q1. Line 341: "We sample 50 questions from the benchmark datasets with low baseline performance under GPT-4o." What are the benchmark datasets? Also is GPT-4o used as the main model in subsequent experiments? This is very confusing since the dataset and model for this section is not introduced.
* Q2. Table 3: Help me understand why the search space is 27 here, but 1296 in the previous section?
* Q3. Table 3: Help me understand how the coverage is computed. E.g., how do you get 21.43%? I was expecting this to be 5 divided by 27 but this seems to be computed differently?
* Q4. Figure 4: How should I interpret this figure? It seems that all methods are having similar accuracies and it's hard to determine which one is the best. Hence it's hard to conclude that PSAO is being effective here. Could the author further explain what's the takeaway here?

---
Suggested reference: https://arxiv.org/abs/2203.07281

---

### Official Review · Reviewer_x4Am · 2025-11-01

**Soundness:** 2
**Presentation:** 2
**Contribution:** 3
**Rating:** 4
**Confidence:** 3

**Summary:**

The paper presents a prompt optimization framework called PSAO, which can be used to optimize existing prompts. The Prompt Semantic Alignment and Optimization (PSAO) framework generates prompts that are more consistent with human reasoning and more readable to humans. PSAO provides soft guidance to large language models (LLMs) without altering the original semantics of the prompt. It is model-agnostic and introduces only negligible computational overhead. PSAO employs prompt segmentation and annotation to identify the most effective way to segment and label the prompt for optimization. The purposed PSAO algorithm is evaluated on randomly selected questions from GSM8K, MMLU, Mutli-Arith, Big-Bench-Hard and AQuA dataset using GPT-4o model and Gemini-2-Flash model.

**Strengths:**

1.  The paper presents a framework to improve existing prompts without making significant changes to their semantics or structure.
2. It provides a theoretical framework for the proposed  prompt optimization algorithm for which give the interpretable prompt.

**Weaknesses:**

1. The experimental results shown in **Table 2** should compare different level of segmentation under similar configurations such as brackets and position and prompt setting.

2. The paper does not include details and analysis of problems which are sampled from the dataset such as GSM8k, MMLU etc.

3. There are insufficient experiments to validate the theoretical proofs such as the theorem 3 which requires the different level of segmentation with fixed conditions.

**Questions:**

1. In **Figure 4**, it is unclear which annotation and system prompt were used for the experiments.
2. **Figure 3:** The two baselines show different performance scores. Do they represent the same baseline? What is the impact of the brackets and prompt settings on the average accuracy with the same segmentations? Why is the comparison not performed under the same prompt settings for each segmentation?
3. In **line 340**, the authors refer to 50 questions, but the sample questions are missing.
4. Regarding the **PSAO algorithm**, why was the comparison not conducted with the original prompt? Isn’t this necessary to validate Theorem1, where $Q(\mathcal{M}(P_{S*, A*}))$ $\ge$ $Q(\mathcal{M}(P))$?
5. How does the PSAO algorithm capture different levels of segmentation, since **line 9** in _Algorithm 1_ seems to generate only one kind of segmentation?
6. Does the accuracy always increase with finer levels of segmentation? What guarantees that it will improve with finer segmentation?
7. Isn’t the **GSM8K** dataset relatively easy for a frontier model like _GPT-4o_?

---

### Note · Authors · 2026-01-12

I have read and agree with the venue's withdrawal policy on behalf of myself and my co-authors.